# Giant ice rings in Southern Baikal: multi-satellite data help to study ice cover dynamics and eddies under ice

Alexei V. Kouraev[1,2], Elena A. Zakharova[3,4], Andrey G. Kostianoy[5,6], Mikhail N. Shimaraev[7], Lev V. Desinov[8], Evgeny A. Petrov[9], Nicholas M.J. Hall[1], Frédérique Rémy[1], Andrey Ya. Suknev[10]

[1]LEGOS, Université de Toulouse, CNES, CNRS, IRD, UPS Toulouse, France

[2]Tomsk State University, Tomsk, Russia

[3]EOLA, Toulouse, France

[4]Institute of Water Problems, Russian Academy of Sciences, Moscow, Russia

[5]P.P. Shirshov Institute of Oceanology, Russian Academy of Sciences, Moscow, Russia

[6]S.Yu. Witte Moscow University, Moscow, Russia

[7]Limnological Institute, Siberian Branch of the Russian Academy of Sciences, Irkutsk, Russia

[8]Institute of Geography, Russian Academy of Sciences, Moscow, Russia

[9]Baikal Museum of the Irkutsk Scientific Center of the Russian Academy of Sciences, Listvyanka Village, Irkutsk Region, Russia

[10]Great Baikal Trail (GBT) Buryatiya, Ulan-Ude, Russia

*Correspondence to*: Alexei V. Kouraev (alexei.kouraev@gmail.com)

**Abstract.** Ice cover on lakes is subject to atmospheric forcing from above and the influence of water dynamics and heat flux from below. One characteristic example of these influences in some large lakes, such as Lake Baikal in Russia, are the giant

ice rings and the associated eddies under the ice cover. In April 2020 a giant ice ring appeared in Southern Baikal and a lens-like eddy was detected below the ice. We analysed the temporal changes of ice cover using satellite images from multiple satellite missions - MODIS on Terra and Aqua, Sentinel-1 SAR, Sentinel 2 MSI, Landsat-8, PlanetScope, satellite photography from the International Space Station, and radar altimetry data from Jason-3. Satellite imagery and meteorological data show an unusual temporal changes of ice colour in April 2020, which was explained by water

infiltration into the ice followed by the competing influences of cold air from above and the warm eddy below the ice. Tracking of ice floe displacement also makes it possible to estimate eddy currents and their influence on the upper water layer. Multi-satellite data contribute to a better understanding of the development of ice cover in the presence of eddies, the role of eddies in horizontal and vertical heat and mass exchange and their impact on the chemistry and biology of the lakes and on human activity.


**Introduction**

Lakes can be viewed as an integrator of climate processes, and also strong indicator of climate change. This dual property stems from the way in which lakes respond to regional and global climate variation. Changing climatic or weather conditions affect the state and variability of natural parameters in lakes, such as temperature and salinity, stratification, currents, frontal zones, upwellings, eddies and gyres, ice and snow conditions, water level and biodiversity. Lakes are part of the Global Climate Observing System (GCOS) Essential Climate Variables (ECVs) (GCOS web site, 2021). Ice cover, together with water temperature and colour is among the key parameters for the GCOS' Global Terrestrial Network for Lakes (GTN-L) (GCOS terrestrial network web site, 2021).

Ice cover modulates heat and mass exchange between lake and atmosphere and affects physical, chemical and biological processes in the lakes (f.ex. Kirillin et al., 2012; Bouffard and Wuest, 2019) . The state of the ice is also important for establishing transport on ice, for fishing activities and tourism (Prowse et al., 2011; Vincent et al., 2012). Ice cover is in constant changes from its formation to complete melting, due to interaction with the atmosphere (heat and wind forcing) above and the influence of the water column (heat and water dynamics) below. One characteristic example of such interaction in some large and deep lakes is giant ice rings and their associated eddies under the ice cover.

Giant ice rings, most often observed in Lake Baikal, are a beautiful and not yet completely understood natural phenomenon. They are rings of dark ice with a typical diameter or 5-7 km. They appear on the ice cover of some large lakes in a seemingly unpredictable manner from year to year. The ice is thinner and appears darker in the ring region, and in the center and outside the ring the ice is thicker and looks white like the surrounding undisturbed ice cover (Granin et al., 2015, 2018; Kouraev et al., 2016, 2018, 2019). Due to their large size one of the best ways to observe and analyse ice rings is from satellite imagery.

Giant ice rings were first observed in various places in Lake Baikal (Granin et al. 2005, 2008, Kouraev et al., 2016) but later we have found them in two other lakes - Lake Hovsgol in Mongolia and Lake Teletskoye in Altai, Russia. The total number of ice rings detected from satellite imagery is now close to 60,with the earliest documented ice rings in 1969 (Kouraev et al., 2019).

Ice rings have attracted the interest of both scientists and the general public, and several hypotheses have been put forward to explain the appearance of this strange phenomenon on the ice cover (see Kouraev et al., 2016 for description). The current consensus is that ice rings are a surface manifestation (on the ice cover) of heat fluxes produced by eddies under the ice, although proposed mechanisms differ for the generation of these eddies.

In-situ observations of water structure beneath the ice rings in Southern (2009) and Middle (2013) Baikal (Granin et al., 2015, 2018) show the presence of an anticyclonic (clockwise) vortex. The authors explain the creation of this vortex by upwelling of deep waters that can be associated with the rise of methane gas hydrates. The first clear physical explanation of this phenomenon appeared after a comparison of satellite imagery of an ice ring with in-situ measurements of vertical structure of water temperature and calculation of the density field (Kouraev et al., 2016). Our hydrographic surveys in the region of ice rings in Lake Baikal (2012-2020) and in Lake Hovsgol (2015) have shown the presence of warm lens-like (double-convex form) intrathermocline eddies beneath the ice cover  (Kouraev et al., 2016, 2019), which are similar to the well known Mediterranean eddies (Meddies) in the North Atlantic Ocean (Kostianoy and Belkin, 1989). CTD (conductivity, temperature and depth sensor) casts have shown that these eddies exist before and continue to exist during ice ring appearance and development. They rotate in a clockwise direction and make a complete rotation around its vertical axis in about 3 days.

Various in-situ measurements have shown that currents in the center of the eddies are absent or weak. They are strongest at the eddy boundary and it is here that increased heat exchange between ice and water leads to ice melting from below and the formation of rings with thinner and darker ice, and not of circles (or round patches), as one might expect.

Analysis of thermal satellite imagery before the formation of an ice ring near Cape Nizhneye Izgolovye (Middle Baikal) in 2016 shows that the eddy was formed in ice-free conditions in late autumn 2015 by an outflow of water from the Barguzin Bay. So in this case the main driver of eddy generation was the wind-induced movement of colder and lighter water (due to the temperature/density relationship for freshwater in the range from 0 to $4^0$C) in association with the coastline shape (Kouraev et al., 2019). We suggest that this mechanism is typical for the eddies in this region and potentially for many other eddies that eventually may generate ice rings.

The typical shape of an intrathermocline eddy in Lake Baikal can be described as follows (Kouraev et al., 2019, see also Fig. 1). It has a double-convex shape with a neutral layer located at about 45-50 m depth (depth of thermocline). The upper part presents a dome-like rising of isotherms (and isopycnals), that displaces laterally the cold well-mixed water that is typical for neighbouring regions under the ice cover. The lower part of the eddy is bigger and deeper in extent than the upper part due to a much weaker vertical stratification of temperature (and density) in this layer and presents a downward inclination of isotherms (isopycnals) extending to depths exceeding 200 m. While the ice ring size is comparable to the upper dome size, the area affected by the eddy in the neutral layer is larger, reaching 10-12 km in diameter (Fig.1).

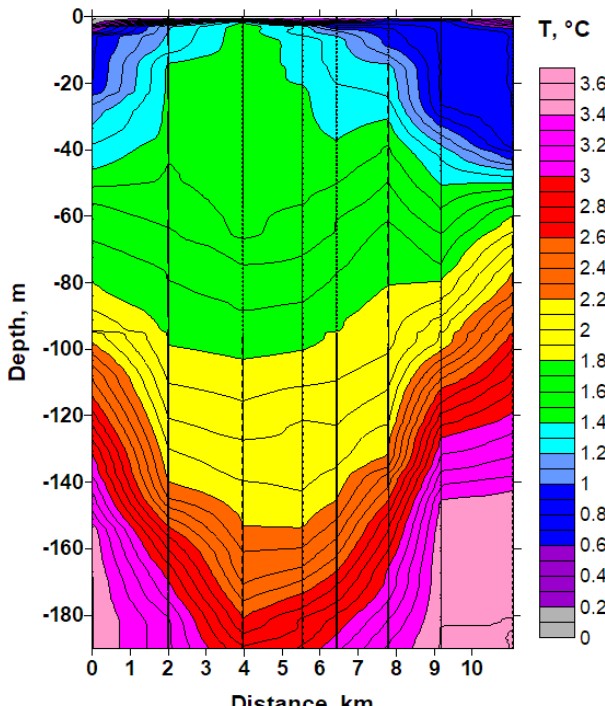

**Fig 1. Vertical section of water temperature (°C) below the ice, 28-30 March 2018, across the eddy in the region of Cape Nizhneye Izgolovye (Middle Baikal). X axis - distance along the transect. Eddy center is located approximately at 4-5 km position. Vertical lines - station positions. This eddy led to the formation of a giant ice ring in late April 2018 (http://icerings.org/news_en.htm).**

In April 2020 another giant ice ring appeared in Southern Baikal. Earlier in situ observations of water structure in this region made on 3-4 April 2020 by researchers from the Limnological Institute in Irkutsk (Russia) revealed an anticyclonic eddy (Zyryanov et al., 2020). Interestingly, the distribution of temperature reveals not a «classical» anticyclonic oceanic eddy with a maximum of the orbital velocity on the sea surface, but an intrathermocline lens-like eddy with the structure described above.

While the formation mechanism for this eddy is not clear, in this paper we would like to address how changes in ice cover can provide new information on the eddy itself. To start with, this case presented  quite an unusual development for an ice ring as seen from the satellite imagery (Figure 2). Typically once an ice rings appears it gets darker and well developed, and then the ice breaks up inside the ring (Kouraev et al., 2016), but there is no significant change in ice appearance outside the eddy nor sudden changes of the ice ring size. However in 2020 it was quite different. For most of April the satellite imagery showed a white surface with the ice ring present and not changing in size. This was followed by sudden changes in ice colour: first the ice rapidly turned very dark for a couple of days, and then the ice ring appeared again. The emerging ice ring was much larger with a sharp contrast between white and dark regions (Figure 2). This unprecedented temporal development of ice is puzzling.

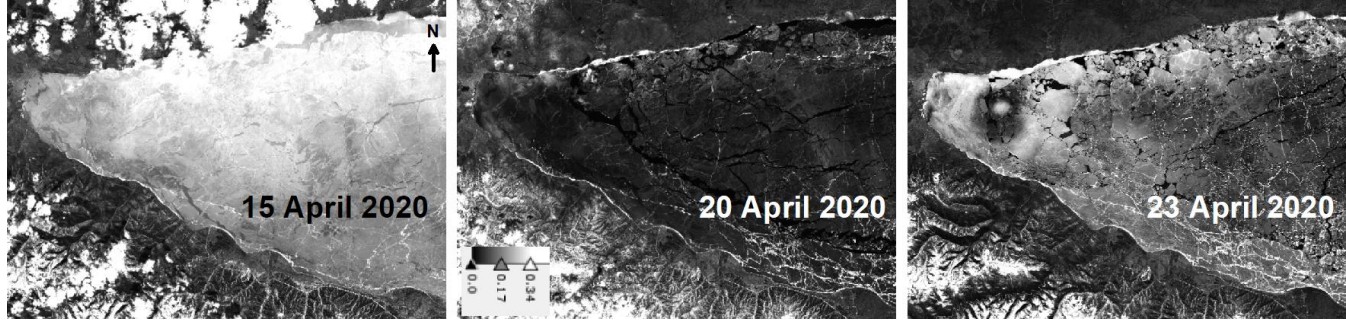

**Figure 2. A sequence of Sentinel-2 images (red band, Level 2A - bottom-of-atmosphere reflectance with atmospheric correction) for Southern Baikal on 15, 20 and 23 April 2020. Colour scheme is the same for the three images. Ice ring that looks like fish eye is located in the westernmost part of Southern Baikal. Dark aspect of ice on 20 April is related to very low reflectance and its causes are explained in Section 3.1. Projection UTM 48N**

Multitemporal satellite data in various parts of the electromagnetic spectra together with meteorological data may reveal some of the physical processes behind these changes. Various available high-resolution satellite imagery (Landsat, Sentinel etc) for the period of observation of this ice ring showed the various stages of ice ring development but the time between available images made understanding difficult. Daily MODIS imagery is potentially useful but has low spatial resolution. Adding daily high-resolution PlanetScope satellite images to the analysis helps to complete the picture.

In this paper we first identify the types of satellite data analysed and describe the geographical location of the study. Then we analyse ice cover changes and metamorphism in the context of meteorological data, and of what we currently know about ice rings and eddies. Finally we will demonstrate how multi-satellite imagery with high temporal frequency and high spatial resolution can help to monitor ice displacement and to reveal the size and impact of the underwater eddy.

## 1. Features of the ice cover and satellite data used

We have used various satellite data to analyse changes in ice cover and first we briefly describe various types of Lake Baikal ice cover (Sokol'nikov, 1960; Verbolov et al., 1965; Atlas of Lake Baikal, 1993, Shimaraev and Verbolov, 1993; Kouraev et al., 2006; 2008; 2016; 2019; Rusinek et al., 2012). Ice freeze-up in calm conditions leads to the formation of crystalline ice (black ice). Due to large latitudinal extent (about 4° latitude) freezing of Lake Baikal is gradual from north to south and can take up to one month. Strong winds during freeze-up can lead to the break-up of newly formed ice, formation of polynyas, leads, small and large drifting ice floes of various thickness and size, pancake ice, ice ridging and rafting, and the formation of hummocked ice with ice floe thickness varying from centimeters to tens of centimeters and hummock height from several centimeters to several meters. Once the whole lake surface is frozen ice grows downwards as crystalline ice and may reach a thickness of 100-120 cm. For the rest of the winter the ice surface will reflect the ice conditions at early freezing time.

During winter ice continues to fracture as a result of thermal inhomogeneity and wind action. Ice contraction lead to the

145 formation of leads that quickly refreeze as crystalline ice, and ice expansion lead to hummocking. Resulting cracks form a complex spatial pattern, and may extend for several hundreds of meters and sometimes for several tens of kilometers.

Snow cover on the ice of Lake Baikal is very thin due to low precipitation and the influence of the wind that blows snow away to the coast. In some regions such as Middle Baikal the wind can completely clear large extents of ice, polishing it to

150 perfection with drifting snowflakes. People sometimes refer to Lake Baikal as a giant skating rink and the ice is one of the main tourist attractions in winter.

Ice cover on Lake Baikal thus presents a heterogeneous pattern changing in space and in time. To analyse ice cover changes we have used various sources of remote sensing data in different parts of the electromagnetic spectrum. Before presenting

them in detail we would like to broadly describe the main groups of data, and what kind of signal satellites receive from different types of ice and snow and from open water (Table 1). The main group is the satellite images in the visible and near-infrared (NIR) range. For some satellites this can also be complemented by the short-wave infrared (SWIR) range. Images show how much energy is reflected by a surface and then received by a satellite. In this paper low reflectance (low energy received by a satellite) is coloured black, high reflectance as white, and intermediate values - by different shades of

grey.

Without going into details about different reflection properties of water and ice in each band, and also not discussing specific cases of acquisition geometry (such as sun glint) we may say that for visible, NIR and SWIR ranges the following gradation can be broadly formulated. Water (large areas of open water or leads) has very low reflectance. Smooth and black crystalline

ice also has low reflectance, but higher than that of water. Rough ice surface, hummocks and expansion cracks have higher reflectance, and metamorphised ice (see section 3.1) higher still. The highest reflectance values are typically from snow-covered ice and clouds.

**Table 1. Satellite data used in this study, their spatial resolution (meter, numerical values) in various parts of the spectrum, and**
170 **intensity of signal on satellite images/data for various surface types.**

| Ranges | Satellite | Aqua/Terra | Sentinel-2 | Landsat--8 | PlanetScope | ISS | Sentinel-1 SAR | Jason-3 | Surface types and intensity of their signal on satellite data |
|---|---|---|---|---|---|---|---|---|---|
| **Reflectance** | | | | | | | | | Very low signal: open water, leads, water-laden ice |
| Visible | | 250 | 10 | 30 | 3 | Var | | | Low signal: Smooth crystalline ice |
| Near infra-red (NIR) | | 500 | 10 | 30 | 3 | | | | Medium signal: rough or hummocked ice, cracks, |

| | | | | | | | | |
|---|---|---|---|---|---|---|---|---|
| Short-wave infra-red (SWIR) | 1000 | 20 | 30 | | | | | metamorphised ice<br>High signal: snow on ice, clouds |
| **Emissivity** | | | | | | | | |
| Thermal infra-red (TIR) | | | 80 | | | | | Low signal: cold surfaces<br>High signal - warm surfaces |
| **Microwave radar** | | | | | | | | |
| Side-looking imaging radar | | | | | | 5 x 20 | | Low signal: open water, smooth ice<br>High signal: rough ice |
| Nadir-looking altimetric radar | | | | | | | 290 | Low signal: rough ice, thick ice, rough water<br>High signal: smooth ice, calm open water |

## 1.1. Satellite imagery

To analyse day-to-day changes in ice cover on large spatial scales we have used MODIS imagery (Moderate Resolution Imaging Spectroradiometer, onboard Terra and Aqua satellites) that has 250 m spatial resolution in the visible range. Both Terra and Aqua provide daily images covering the whole of Lake Baikal. More detailed analysis was done using high-resolution data from Landsat-8 and Sentinel-2 satellites. Landsat 8 OLI (Operational Land Imager) has 15 m spatial resolution in panchromatic and 30 m in the visible, near (NIR) and short-wave (SWIR) infrared ranges; the satellite has a 16-day repeat cycle. We have also used images from Landsat 8 TIRS (Thermal Infrared Sensor) instrument that has 100 m spatial resolution. These images in thermal infrared (TIR) show not reflectance but emission and are closely linked to the temperature of a surface. Sentinel-2 MSI (Multi-Spectral Instrument) has 10 m spatial resolution in the visible and NIR, and 20 m in SWIR ranges. A constellation of two satellites (Sentinel-2A and -2B) provides temporal resolution of 2-3 days for Lake Baikal. MODIS, Landsat-8 and Sentinel-2 have sun-synchronous orbits and they revisit each place at the same local time (late morning over Lake Baikal).

We also used radar images from Sentinel-1 SAR (Synthetic Aperture Radar) in Level 1 GRDH Interferometric Wide (IW) swath mode in VV and VH polarisations. These SAR images have spatial resolution 5 by 20 m and they were terrain corrected and processed with ESA SNAP software. These are active microwave radar observations. The SAR is an imaging instrument. Satellite radar emits a signal and samples the received backscatter (echo) expressed in dB. For side-looking SAR images smooth surfaces, such as calm water, provide a very low echo (dark colour on images), and rough surfaces, such as hummocked ice etc, a high echo (light colour on images) (Table 1).

Gaps in time for availability of high-resolution imagery and the presence of cloud cover present some problems. Key moments in ice cover development in April 2020 are missing. But significant improvement was achieved by including PlanetScope imagery (Planet Team, 2017) into the analysis. PlanetScope is a constellation of approximately 130 cubesats that provides daily images in the visible and NIR range with 3 m spatial resolution. PlanetScope scenes are also taken in late

morning local time. Depending on the date there are either some gaps for the region of study or some areas are seen by different PlanetScope satellites with several minutes between scenes.

## 1.2. Space photography

We have also analysed satellite photography from the International Space Station (ISS) from the "Uragan" program ("Hurricane" in Russian). This program is the continuation of Earth surface monitoring from orbit stations initiated in 1976 from the "Salyut" and then "Mir" stations (Evans et al., 2000). The 'Uragan' program started on 1 January 2001 on the Russian orbital segment during the first expedition to the ISS, and is ongoing. "Uragan" has 20-30 tasks for monitoring natural processes, disasters and catastrophes. Cryospheric processes (glaciers, ice cover etc) are among the priorities, and monitoring of giant ice rings is one of the tasks. The high resolution photos of ice rings in 2009 shown in Figure 3 were taken on 5 April (32R3717, focal length 800 mm) and on 25 April (32R8494, focal length 300 mm) from the 400 km altitude. Unfortunately in April 2020 astronauts were not able to take photos of the ice ring due to the their acclimatisation regime.

## 1.3 Radar altimetry

Additional information was obtained by satellite radar altimetry. While the main mission of radar altimeters is the monitoring of water level over the ocean or large water bodies, the return signal also provides valuable information on the state of ice-covered or open water surfaces. We have used data from the Jason-3 satellite, track 79 passing across Kultuk Bay (see Fig 1), cycles 141-157 (9 December 2019 - 16 May 2020) for the analysis. We used data from the nadir-looking radar altimeter operating in the Ku band (13.6 GHz), with the backscatter parameter processed with the Ice retracker. The backscatter coefficient is the ratio of the power reflected from the surface to the incident power emitted by the onboard radar altimeter, expressed in decibels (dB). A rough water surface typically has a low backscatter coefficient, while over ice cover it is high. The satellite orbit is non sun-synchronous. The repeat period is slightly less than 10 days along the same ground tracks. Radar altimetry does not provide images, but point measurements along the satellite track: 20 Hz data provide an along-track ground resolution of about 290 m. One of the clear benefits of radar altimetry, as well as SAR imagery, is that they are independent of cloud coverage. Microwave signals penetrate clouds, and they do not need sunlight for observations as the signal is emitted by the satellite itself.

## 2. Kultuk Bay and its ice rings

Southern Baikal, or more precisely its extreme south-western part called Kultuk Bay, is one of the several regions in Lake Baikal where ice rings are relatively common (Figure 3, see also statistics of ice rings in Kouraev et al., 2016). Two other such regions are Cape Krestovskiy and Cape Nizhneye Izgolovye in Middle Baikal. Kultuk Bay is surrounded by mountains on the northern and southern coasts, and in the western part it communicates with the 190-km long Tunka Valley, which is

oriented mostly west-east. Strong and persistent wind from Tunka Valley affects most parts of Lake Baikal. All around Lake Baikal people call this wind "Kultuk" in reference to its origins.

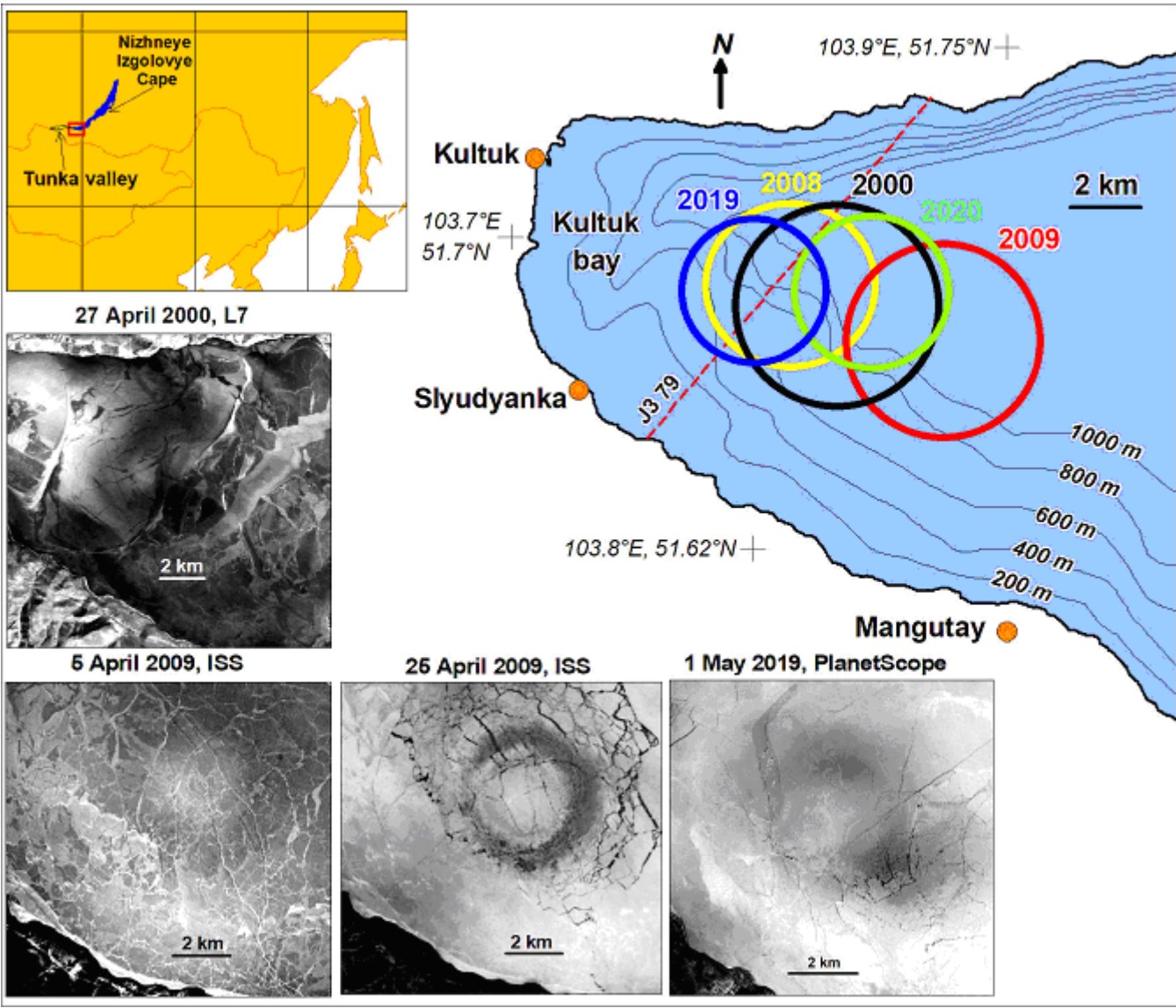

Figure 3. Overview map of the southern part of Lake Baikal with bathymetry and position of the giant ice rings in different years (coloured circles denote outer limits of ice rings) and some examples of ice rings from satellite imagery: 27 April 2000 (Landsat 7, band 4 near-infrared), 5 and 25 April 2009 – photography from ISS (red channel, georeferenced and reprojected), and 1 May 2019 (PlanetScope, near infrared band). Red dashed line marked "J3 79" - Jason-3 track No 79 over Lake Baikal. Projection: UTM Zone 48 North (WGS84).

The bottom of Kultuk Bay forms the western part of an abyssal plain with depth of more than 1500 m that occupies the main part of Southern Baikal. Near the coast this abyssal plain has boundaries with steep slopes. On the southern coast the

inclination of these slopes is about 10° (isobath 1000 m is located at about 5.5 km from the coast). The northern slopes are extremely steep: inclination 30-38° (in some cases the 1000 m isobath is just 1.3 km away from the coast).

One may safely say that it is Kultuk Bay that made giant ice rings known worldwide. After a photo of an ice ring in 2009
(see Fig 3, image from 25 April 2009) taken by astronauts from the International Space Station was posted at NASA Earth Observatory web site ("Circles in thin ice", 2009) and then on other media sources, ice rings became an internationally known phenomenon. This spurred several scientific publications and initiated wider scientific research, including our own studies.

So far there are five documented cases of observations of ice rings on satellite images and space photography in Kultuk Bay (Figure 3, Table 2). Their average diameter is 4.7 km which is slightly smaller than many other ice rings in Lake Baikal (Kouraev et al., 2016, 2019) as the development of eddies is probably limited by the size and shape of Kultuk Bay. The duration of their manifestation on the ice cover was 15-17 days with the exception of 2009, when ice ring was visible one week longer. The last sighting of ice rings is just a few days before ice break-up and melt, a typical feature for most ice rings
detected in Lake Baikal. The first observation of an ice ring in Kultuk Bay was in 2000 and this is probably determined by the paucity of available satellite imagery before 2000. We have also discovered an as yet undocumented ice ring in Kultuk Bay in May 2019 from Sentinel-2 imagery.

**Table 2. Inventory of all ice rings detected so far in the Kultuk Bay and their characteristics. For the method used to define ice**
**rings and the previous inventory see Kouraev et al. (2016, 2019).**

| Year (end of winter) | Diameter, km | Lon E | Lat N | First seen[a] | Last seen[a] | Observation, days[b] | Open water date | Typical depth, m |
|---|---|---|---|---|---|---|---|---|
| 2000 | 5.6 | 103.83 | 51.68 | 27/04 | 27/04 | (1) | | 1000 |
| 2008 | 4.4 | 103.81 | 51.69 | 16/04 (1) | 30/04 (3) | 15 | 05/05 | 800 |
| 2009 | 5.2 | 103.88 | 51.67 | 04/04 (3) | 27/04 (2) | 24 | 05/05 | 1000 |
| 2019 | 4 | 103.80 | 51.69 | 15/04 (1) | 01/05 (1) | 17 | 04/05 | 600 |
| 2020 | 4.2 | 103.85 | 51.69 | 08/04 (2) | 24/04 (1) | 17 | 04/05 | 1000 |

[a] Date format is (DD/MM); numbers in brackets - days since last ring-free scene for first ring seen, and days to first ring-free scene after last ring observation, [b] duration is defined as difference between the first observation and the last one. For observation in 2000 based on non-MODIS imagery, duration is put in brackets, meaning "at least X days)", though the ring could have existed longer.


All five rings were located on the southern slope of the abyssal plain, very close to one another, with the distance between their centers less than 5 km. In our previous work (Kouraev et al., 2016, 2019) we first suggested and then documented that lens-like eddies under ice may change their position, in some cases not even leading to the formation of ice rings. We also

suggested that while travelling, eddies may be trapped in extremities of abyssal plains, such as near Cape Nizhneye
Izgolovye. Kultuk Bay with similar bathymetry may be another such place where eddies are trapped.

## 3. Development of ice cover, giant ice ring and eddy in 2020

The combination of multi-satellite imagery and data makes it possible to analyse in detail the development of ice cover in Southern Baikal for winter 2020, with a focus on Kultuk Bay and the region of the giant ice ring observed in April 2020. The first ice floes in Kultuk Bay appeared on 8-9 January and six days later the whole of Southern Baikal was frozen. The
appearance of young (nilas) drifting ice led to a sharp increase of backscatter from 15 to 45 dB on the Jason-3 ground track (Fig. 4).

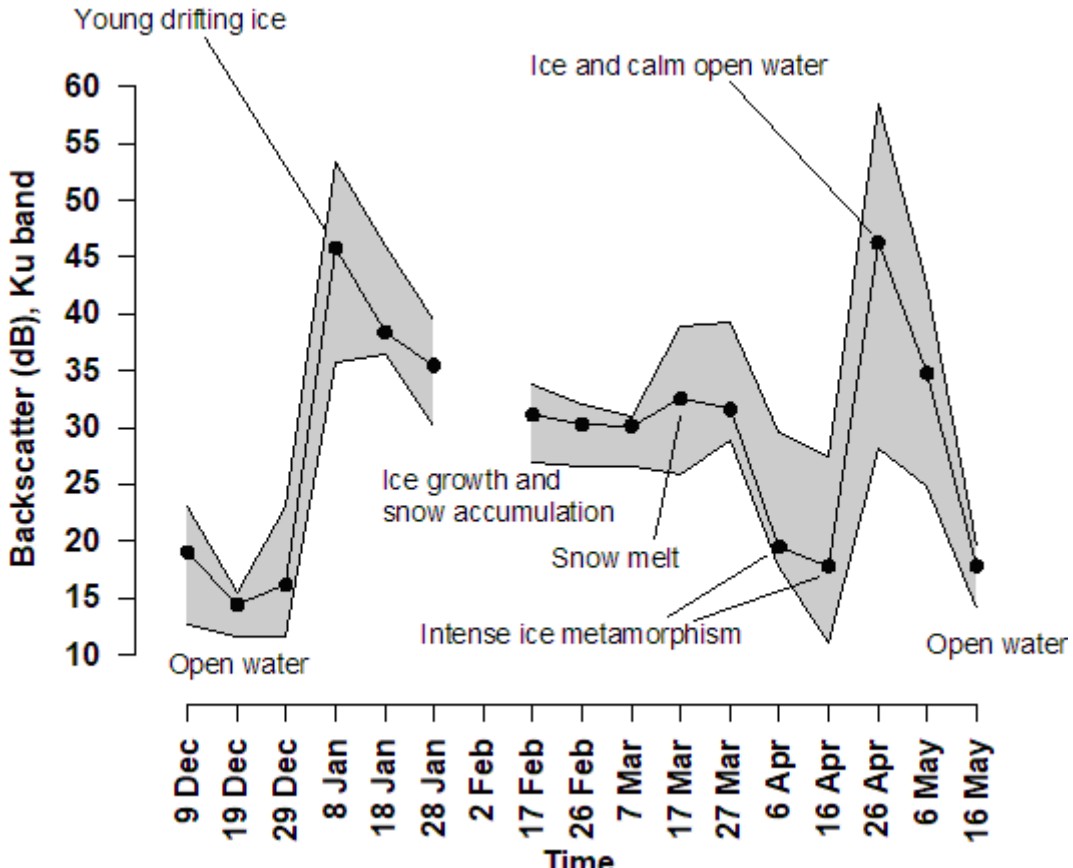

Figure 4. Temporal changes of Jason-3 backscatter coefficient (dB) in Ku band, track 97  in winter 2019/2020. High backscatter
values correspond to high return echo from radar signal. Black line with dots - median values, grey zone - spread between maximum and minimum values.

Up to the end of March 2020 most of the southern part of the lake was snow-covered. A gradual decrease of Jason-3 backscatter down to 29-30 dB by the beginning of March indicates of snow accumulation, ice growth and roughening. Snow disappearance by sublimation in the second half of March exposed a whitish surface of metamorphised ice. This is seen on the visible images and also in a slight (2-3 dB) increase of Jason-3 backscatter.

Snow completely disappeared by 2 April, and on 8 April an ice ring was detectable for the first time. This giant ice ring had a circular shape with an outer diameter of 4.2 km and the width of the dark ring was 0.9 km (Fig 5a). The image for 15 April shows the presence of numerous ice fractures (leads, seen as dark features) in the ice ring itself. They are mostly orthogonal to the ice ring and their length is comparable or slightly longer than ring's width. There are also several longer fractures and leads in the region up to 1 km outside the eastern and north-eastern limits of the ice ring. The width of the leads is 20-30 m. Jason-3 data for the next day (16 April, also shown on Fig 5a) confirms the existence of these leads as areas of smooth and calm open water, with a specular return for the nadir-looking radar altimeter. Although this high backscatter is mixed with lower backscatter from neighbouring ice fields, the resulting echo values are still high - in the range of 20-28 dB.

### 3.1. Ice break-up, displacement and metamorphism.

Ice cover in the southern part of Lake Baikal was stable until 18-19 April 2020. After that date the ice state was affected by several meteorological factors – wind, air temperature and precipitation.

**Wind influence**. According to data from the Kultuk meteorological station, starting from 18 April a constant Kultuk wind from the Tunka Valley blew with an average speed varying between 3 and 8 m/s and gusts up to 14 m/s (Fig 6). On 20 April the wind initiated ice break-up east of the ice ring region (Fig 5b). The next day under continuing westerly winds, the ice in the ring region itself was broken onto several ice floes (Fig 7a, 21 April 2020). A large ice floe "A" with a diameter of about 3 km corresponds to the initial center of the ice ring and eddy. As mentioned in the Introduction, this floe is thicker because eddy currents are weaker in the center of the eddy. During break-up this ice floe moved slightly to the north-west from its initial position. On 21-23 April the wind was weak and variable, so the position of the ice floes did not change much (Fig 7a-c), except for slight compacting to the west between 21 and 22 April 2020.

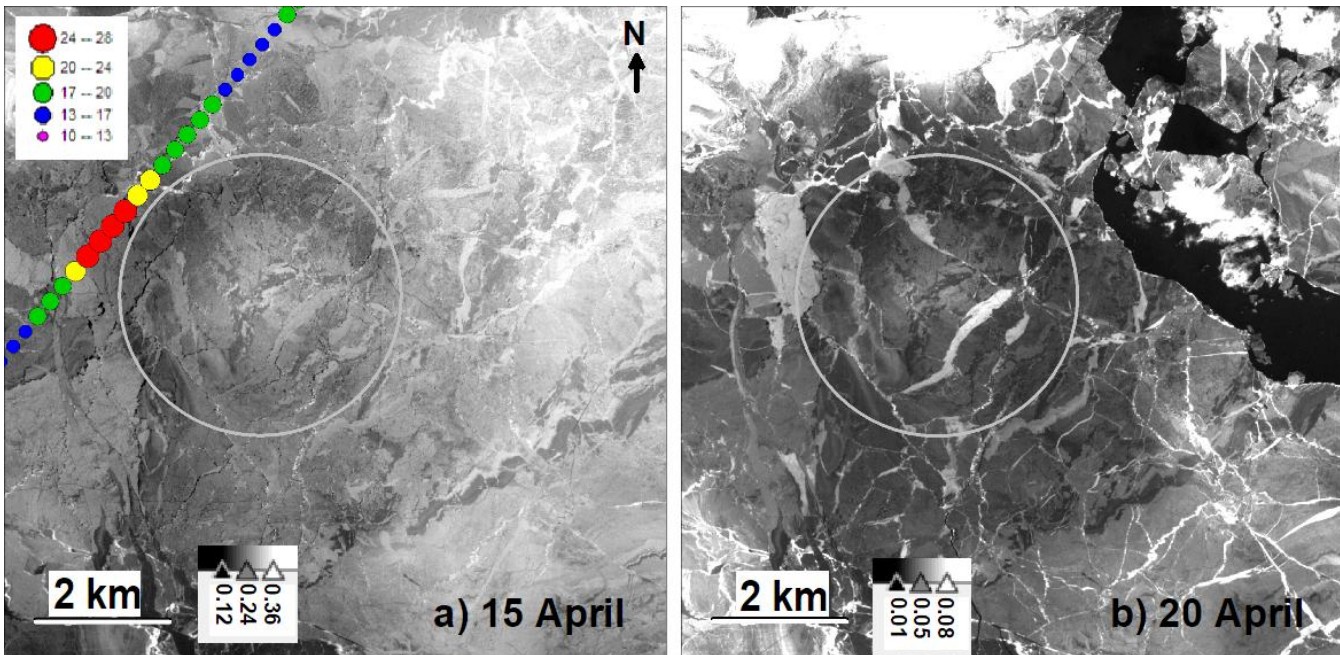

Figure 5. Changes in ice cover in April 2020. on satellite images. a) 15 April and b) 20 April, Sentinel-2B, red band. Grey palette - reflectance units (digital numbers). On image for 15 April are also shown backscatter values (dB, coloured dots) for the Jason-3 track 79, cycle 154 (16 April 2020, 22 h local time, GMT+8h). Grey circle – outer limit of ice ring as defined from image on 15 April. White linear features - cracks, black areas - open water. All satellite images on Figures 5, 7 and 9 are to scale and show exactly the same region. Color stretching is different for each image to enhance the contrast. Projection UTM 48N.

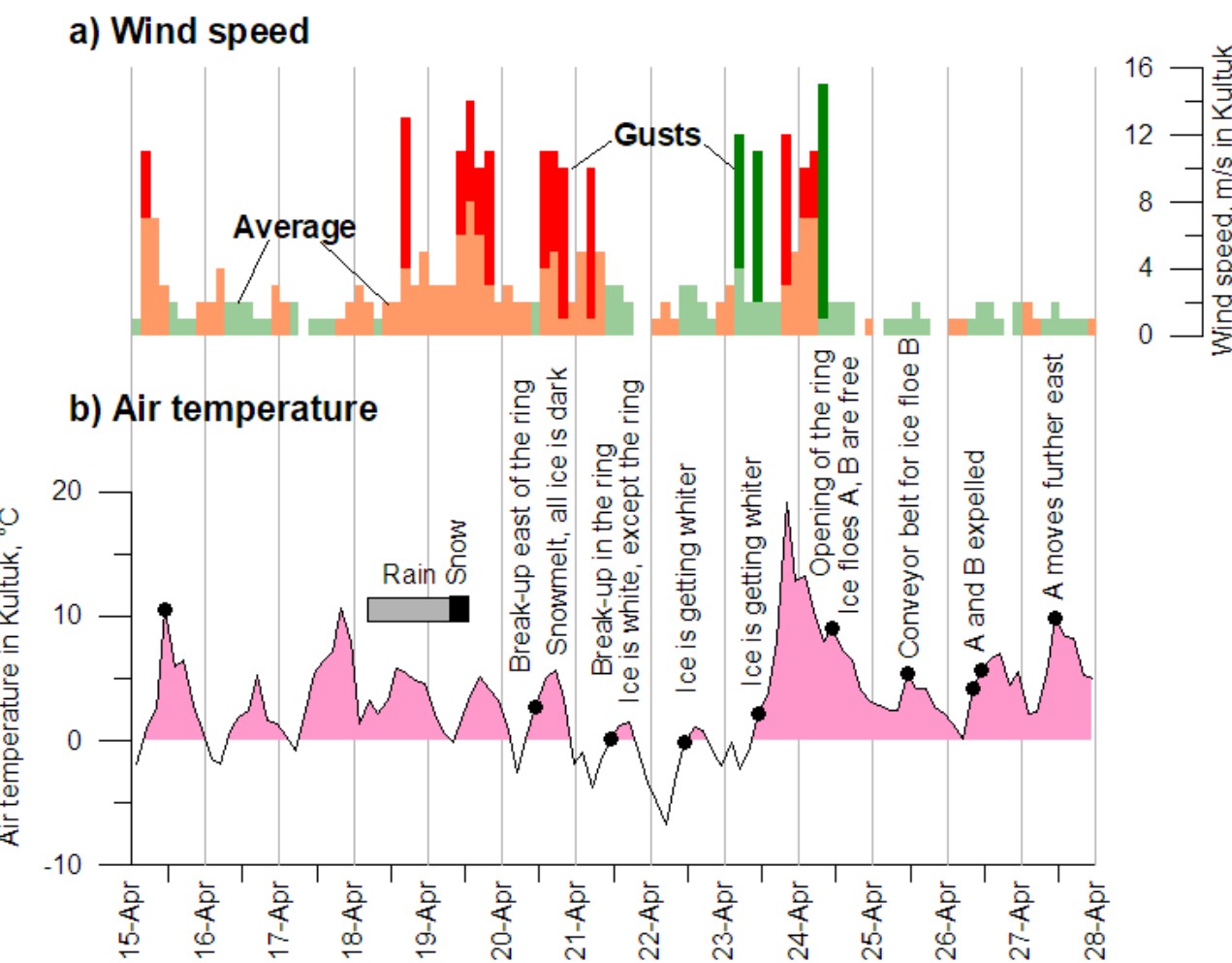

**Figure 6. Temporal changes of a) wind and b) air temperature for 15-28 April 2020 at the Kultuk meteorological station (Russian Hydrometeorological Service). The period is selected to represent meteorological conditions for satellite imagery presented in the paper. Wind speed is coloured as a function of the general direction. Wind coming from the Tunka Valley (Kultuk wind) was classed as wind coming from the direction 292.5°(WNW, main direction of the opening of the valley) with a ± 45° span: 247.5 to 337.5°, or wind between WSW and NNW. Kultuk wind is coloured in peach/red, winds from all other directions – in light/dark green. For each of the two directions we also present two different estimates of wind speed: average wind speed (lighter colour), and maximal gusts (darker colour). Black circles on the air temperature graph – date and time of satellite images discussed in the text.**

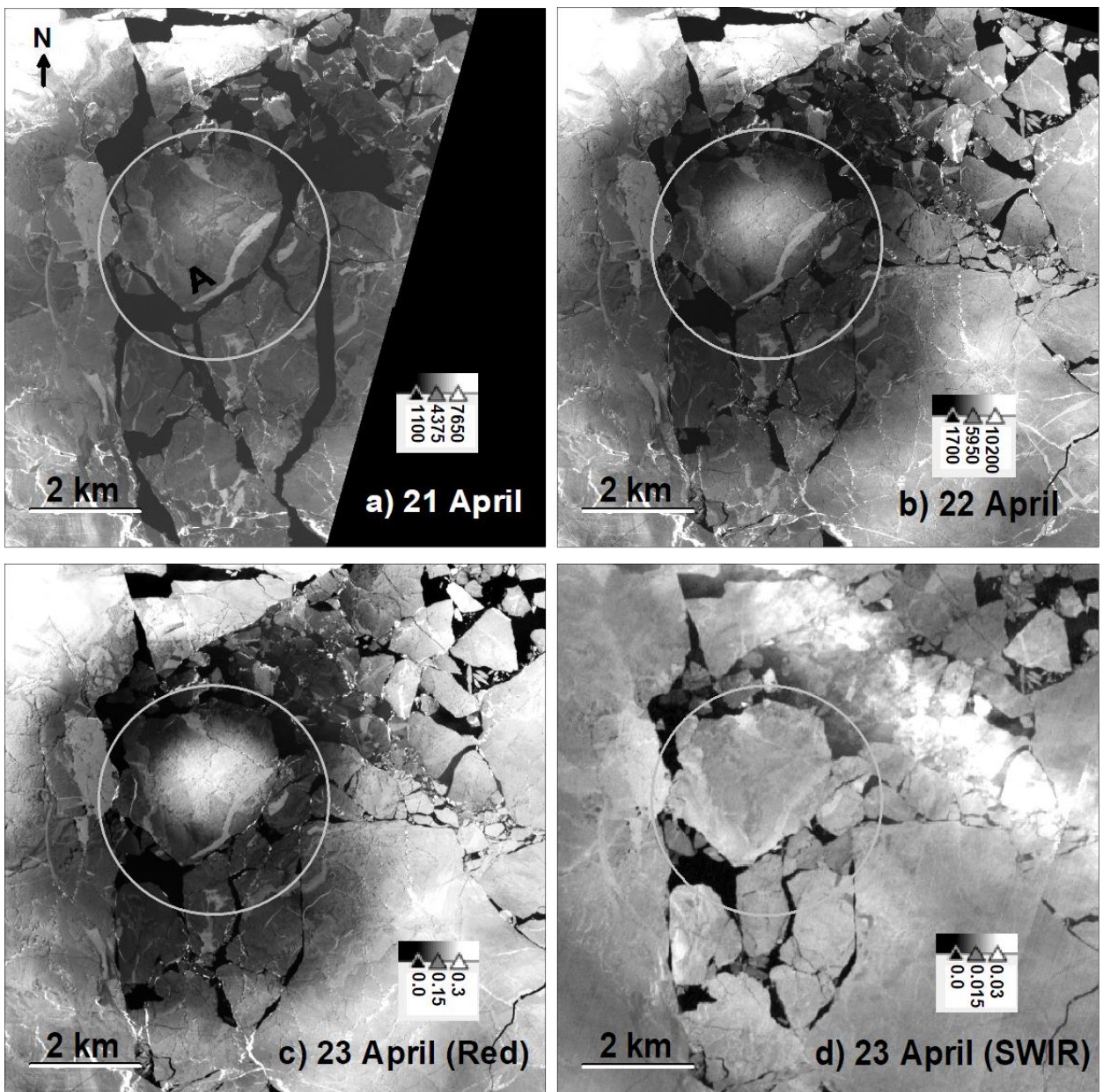

**Figure 7. Same as Fig 5, but a) 21 April and b) 22 April - PlanetScope, red band, c) 23 April, Sentinel-2A, red band, d) same image as (c) but SWIR band. A – large ice floe mentioned in the text. White linear features - cracks, black areas - open water. Color stretching is different for each image to enhance the contrast. Grey palette - reflectance units (digital numbers), specific for each satellite and band. Projection UTM 48N.**

**Ice metamorphism, warm air and precipitation**. At the beginning of April 2020 ice in Southern Baikal had already metamorphosed. Typically under intense solar radiation vertical crystals of columnar ice start to melt from the surface (Sokolnikov, 1959; Obolkina et al., 2000). This process is not specific to Lake Baikal. Such metamorphisation is also seen in

many other lakes. Melting starts at the boundaries of the ice crystals, as the presence of impurities there decreases the

melting point (Ashton, 2007). Ice metamorphism can take different forms, and some of them (Fig 8) can be vertical columns of tiny bubbles, large bubbles near the surface, or channels of air delineating boundaries of columnar ice crystals. In contrast to many shallow boreal lakes, the majority of columns of vertical bubbles for Lake Baikal are not related to methane activity in the sediments but to melting from direct insolation. Large and small channels of air just below the ice surface, called "shakh" in Russian, are the predominant form of metamorphism for Lake Baikal ice. Sometimes the formation of air

channels is quite rapid and can be heard and observed directly (Kouraev et al., 2015). This metamorphism turns dark transparent crystalline ice into white ice and causes high reflectance seen on satellite images in the visible and near-infrared.

The ice metamorphism increases the albedo, reducing the impact of solar radiation and delaying melting. It also alters ice radiometric properties in the microwave range. The radar signal is very sensitive to the size and volume of scatterers in lake

ice, especially air bubbles on side-looking SAR signal (Duguay et al., 2014; Atwood et al., 2015; Gunn et al., 2018). For nadir-looking radar altimetry we have previously documented large temporal changes in backscatter for ENVISAT/RA-2 and SARAL/AltiKa radar altimeters over the Middle Baikal (Kouraev et al., 2015). For Jason-3 observations ice metamorphism led to significant decrease of backscatter on 6 and 16 April (see Fig 4) - down to 15-20 dB. These low values are almost comparable backscatter from rough open water.


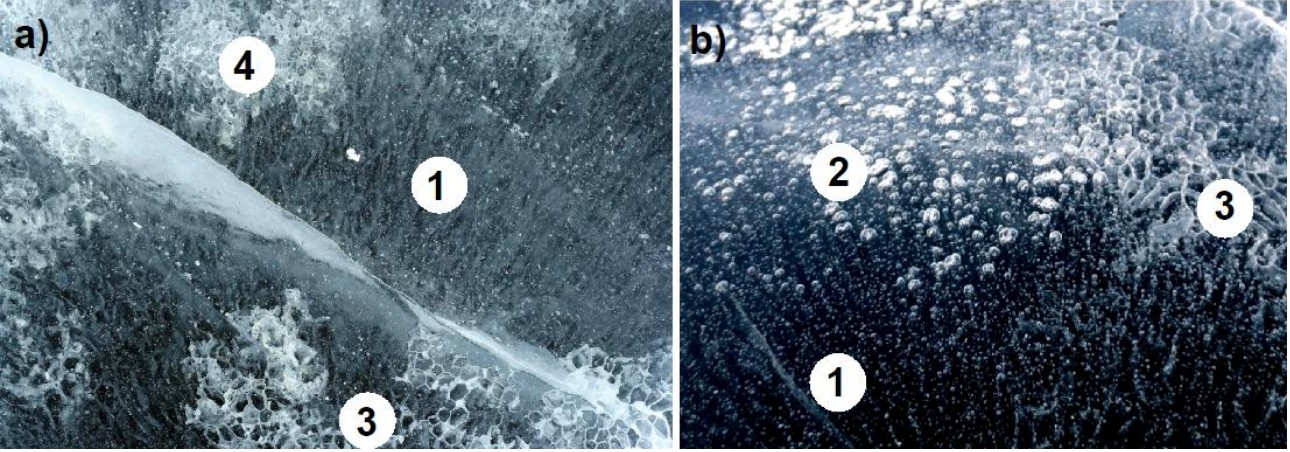

**Figure 8. Different manifestations of ice metamorphism on the ice cover surface. a) Southern Baikal, 10 April 2006 (Photo: E. Petrov) b) Middle Baikal, 3 April 2014 (Photo A. Kouraev). Photo shots were taken while standing on ice (about 1.5 m height). 1 - vertical lines of air bubbles; 2 - large bubbles just below ice surface; 3 - air channels showing boundaries of columnar ice crystals;**

**horizontal size of these crystals is about 2-4 cm; 4 - similar to 3, but upper central parts of columnar ice is also white.**

High values of air temperature of over +10°C during the day and close to 0°C at night were prevalent over six consecutive days from 15 to 20 April 2020 (Fig 6b). At 5 h (local time) in the morning on 18 April light rain showers started and persisted until 8 h 19 April, followed by light snow showers until 14 h on 19 April (also marked on Fig 6b). The air

temperature was positive (up to 5.1°C) during the day and at night briefly decreased to –2.6°C, so rain and snow created a liquid water layer on ice surface. Apparently this water infiltrated the ice surface, filling the cavities that were previously giving a whitish aspect to the ice. As a result the ice surface turned very dark and extremely low reflectance was observed on MODIS (Terra and Aqua), Sentinel-2 (Fig 2) and PlanetScope images on 19 and 20 April 2020 over a large area of Southern Baikal.


**Cold event reveals the eddy**. After another warm day on 20 April 2020, night temperatures fell down to negative values (down to –3.8°C) for the whole night of 20-21 April. As a result starting from 21 April satellite images show very high contrast. There is still very dark ice in the region initially affected by the ring, but ice in the center of ice floe A and elsewhere outside the ice ring region is white. This tendency continued over the next two days. After a cold night 21-22

April (down to –6.7°C) a large area covering most of Southern Baikal became whiter. The white area of the ice floe A got larger and the dark area in the ring region got smaller. The day of 22 April was cold (maximum value 1.1°C) as was the following night (minimal values -2.3°C). Consequently, there was a stark contrast between white ice in the center of ice floe A and in the outside regions, and a dark ice area located in and directly outside the ring region (see Figs. 2 and 7).

The situation on 23 April 2020 was recorded by images from MODIS sensors, Sentinel-2, PlanetScope and Russian Canopus-B ("Remote sensing..", 2020) satellites. In some cases these images were taken by journalists as evidence of a giant methane bubble trapped under ice and just waiting to explode ("Giant gaz bubble..", 2020).

However we now have a more realistic explanation for these images. We have seen that warm weather and rain and snow

eliminated the surface manifestation of earlier ice metamorphism and led to very dark ice on 19-20 April 2020. This is comparable to preparing a clean canvas for drawing a new picture. Then two painters - cold air from above and a warm eddy from below - went to work on a new picture on ice.

Negative air temperature (data from Kultuk meteorological station) likely led to the formation of a thin crust on the ice

surface, turning it whitish again. This is seen on images from 21-23 April 2020 and is not limited to the eddy region but affects a much larger area in Southern Baikal (see also Fig. 2). Below the ice, eddy influence (warmer water below the ice and increased heat exchange due to stronger currents) counteracts the impact of cold air, delaying or cancelling the formation of a white crust. At the periphery where the eddy current is stronger and where the ice is thin or broken, we continue to see dark ice (Fig 7, a-c). This is further confirmed by comparing the Sentinel-2 images for 23 April in different bands. Contrast

between a white center and a dark ring region was observed in three visible bands and one near-infrared band of MSI sensor, but not at all in SWIR (Fig 7d). The SWIR band with its longer wavelength is better than other bands for seeing thin clouds, such as the condensation trail from an airplane in the upper right corner on Fig. 7d. It is also less affected by reflection from

small-scale surface phenomena, such as ice crust, some types of snow etc. As a result we clearly see the distribution of ice floes and fields with different signatures, but no ice ring.

## 3.2. Ice tracking as a mean to assess eddy currents.

During our study of changes in ice cover and metamorphism we collected numerous satellite images, sometimes several per day, for Kultuk Bay in April 2020. This provides an opportunity to assess the speed of the eddy current by analysing ice flow movement. In the early morning of 24 April a Kultuk wind with gusts of up to 11 m/s led to the displacement of ice floes 1.5-2.5 km to the south-east and the opening of the most of the area affected by the eddy. Starting from this time, the displacement of ice floes allows us to monitor and quantify the influence of the eddy by looking at the changing position of ice floes A (discussed earlier) and B (Fig 9).

An elongated ice floe B with size of 2 km by 1.5 km was detached from the ice field west of the ice ring. Under the influence of the Kultuk wind between 23 and 24 April 2020 it was displaced to the region of the eddy's outer boundary, where current speeds are greatest. Then, like a suitcase thrown onto a conveyor belt at the airport, this ice floe was rapidly transported along the eddy boundary. The image from 25 April (Fig 9b) shows that over one day ice floe B was transported for 6 km (average speed 7 cm/s) almost without changing its orientation with respect to the eddy. Two lines of smaller pieces of broken ice floes follow the trail of ice floe B (Fig 9b) indicating curvilinear displacement along the eddy boundary. Then on 26 April when it met an obstacle (ice floe A), ice floe B was expelled a further 2.3 km to the east, out of the eddy region (Fig 9 c,d). As the eddy no longer affected this ice floe, this time displacement occurred without changing orientation to the North.

Ice floe A was located inside the eddy and its various parts were affected differently by eddy currents. As a result this ice floe manifested not so much lateral displacement as clockwise rotation. A thin and 2.3 km-long fish-shaped band of white ice on the surface of ice floe A helps reveal this rotation.

A sequence of images for 24-26 April 2020 (Fig 9) and PlanetScope image for 27 April (not shown) allows us to define the positions of ice floes A and B at different dates (Fig 10) and estimate their rotation (Table 3). Ice floe B, whilst inside the eddy, was rotated 97° clockwise between 24 and 25 April. Ice floe A experienced strong clockwise motion between 24 and 27 April, with a total rotation of 219°. The rotational speed of ice floe A decreased when it left the eddy region, but still continued, possibly due to the to angular momentum. This was not the case for the ice floe B which was on the periphery of the eddy.

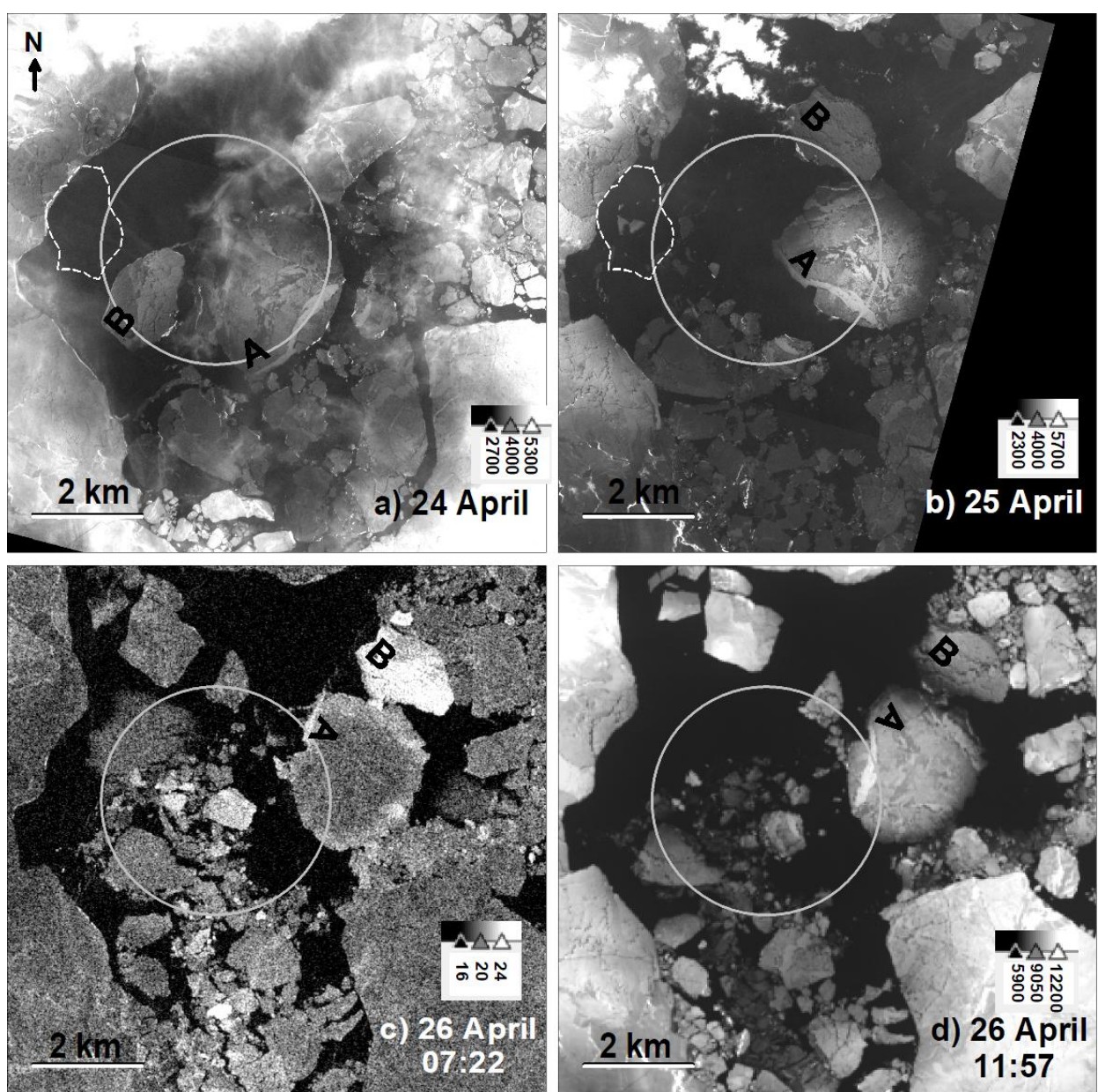

Figure 9. Same as Fig. 7, but f) 24 April and b) 25 April 2020, PlanetScope, red band, c) 26 April 2020, Sentinel-1B SAR, VV polarisation d) 26 April 2020, Landsat-8, red band. A and B – large ice floes mentioned in the text, dotted line on (a) and (b) – position of ice floe B on the 23 April 2020. White linear features - cracks, black areas - open water. All time for satellite images is local time (GMT+8). Color stretching is different for each image to enhance the contrast. Grey palette on a,b and d - reflectance units (digital numbers) specific for each satellite, c - dB values. Projection UTM 48N.

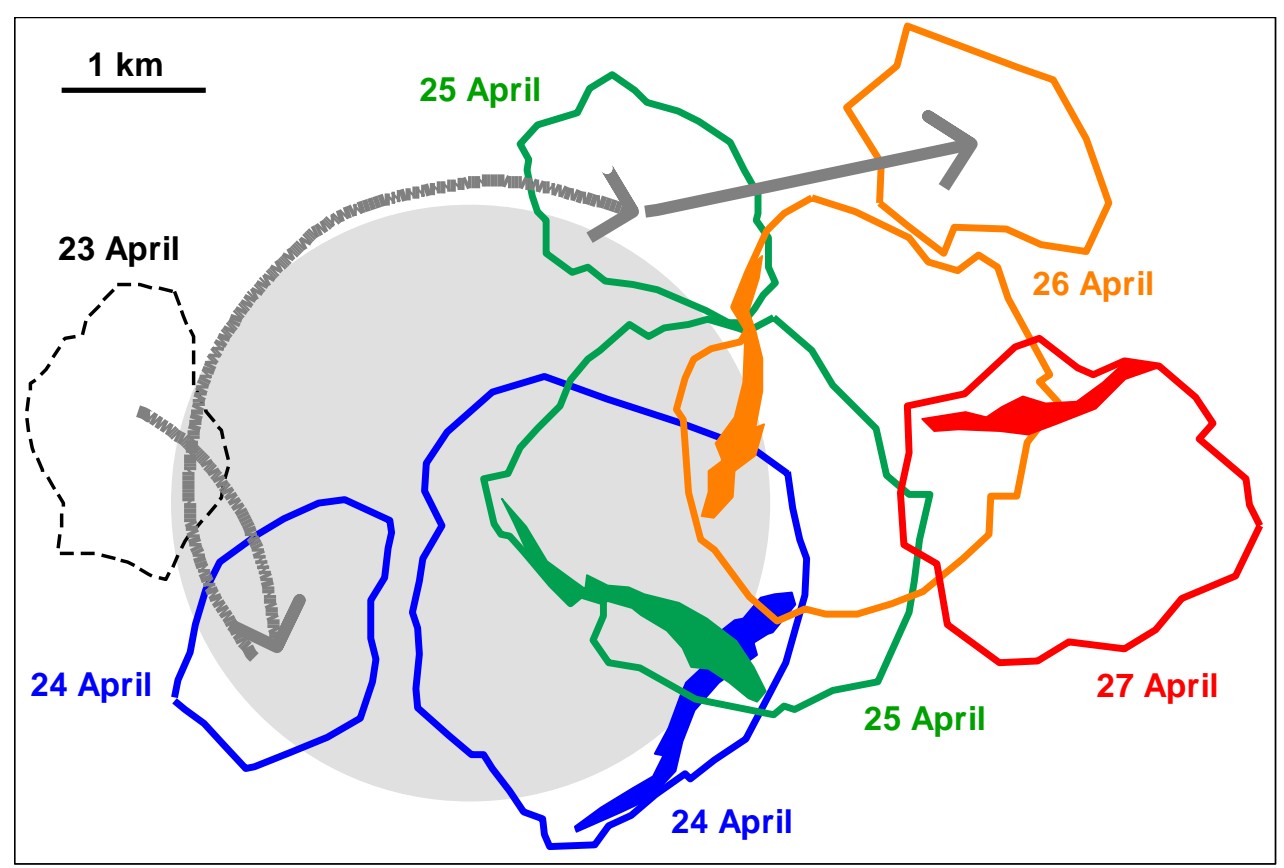

**Figure 10. Schematic representation of position and displacement of ice floes A and B between 23 and 27 April 2020. Grey circle – approximate area affected by eddy, defined from ice ring outer limit on 15 April 2020.**

**Table 3. Angle and angular speed of ice floes A and B in April 2020.**

| Date and local time | Satellite | Angle of ice floe A (deg) | Angular speed (deg/hr) | Angle of ice floe B (deg) | Angular speed (deg/hr) |
|---|---|---|---|---|---|
| 24 April 11:42 | PlanetScope | 0 | | 0 | |
| 25 April 11:47 | PlanetScope | 84 | 3.49 | 97 | 4.02 |
| 26 April 07:22 | Sentinel-1 | 150 | 3.37 | | |
| 26 April 11:57 | Landsat-8 | 156 | 1.31 | | |
| 27 April 11:17 | PlanetScope | 219 | 2.70 | | |
| | | | Average: 3.05 | | |

### Discussions and conclusions

Imagery from multiple satellite missions, meteorological data and knowledge of water dynamics under ice, all taken together enhances the scope of analysis of the development of a giant ice ring and eddy in April 2020 in the Kultuk Bay, Lake Baikal.

**"Redrawing" of the ice ring.** Temporal analysis of ice metamorphism and development helps us to understand and interpret the interplay between two influences. One from above: wind, sun radiation, air temperature, snow and rain and one from below: warm eddy, currents. We have seen that ice reflectance on satellite images changes from white (metamorphised) to uniformly dark and then to a contrasting pattern around the eddy. This was caused first by water infiltration into the ice and then by the competing influences of cold air from above and warm water from below the ice, supplied by the subsurface lens-like eddy.

It is interesting to note that after 20 April 2020 the dark area became larger (7.2 km diameter) than the initial ice ring (4.2 km diameter), although its center did not change much. As ice floe A moved NW from its initial position between 20 and 21 April, and numerous ice floes in the ring region slightly compacted westwards between 21 and 22 April (see Fig. 7), this created the impression that the dark region had an elliptical form.

**The conveyor belt.** Tracking of ice floe displacement also makes it possible to estimate eddy currents and their influence on the upper water layer. We have seen how the eddy transports and expels ice floe B and how is spins ice floe A. This is in agreement with what we know about the spatial distribution of currents in the eddies under ice cover in Lake Baikal (Kouraev et al., 2016, 2019).

Estimation of rotational speed (Table 3) gives an average value of 3.05 °/hr, equivalent to a full rotation in 4.9 days for ice floe A and 4.02°/day or a full rotation in 3.73 days for ice floe B. Our estimations of rotational speed for a similar eddy observed in 2016 near Cape Nizhneye Izgolovye in Middle Baikal (Kouraev et al., 2019) from direct (current loggers) and indirect (temperature loggers) observations indicate a full rotation every 3 days. Rotation speed from ice floes A and B appear close to these values, given that there will inevitably be differences due to the duration of the effect, frictional losses, drag coefficient, wind forcing, etc.

The situation observed in April 2020 in Kultuk Bay is a relatively rare case, when during ice break-up ice rings don't just develop and then disappear, as was the case in 2009 and 2016 for Cape Nizhneye Izgolovye. In such a case large-scale ice break-up allows us to observe how the eddy transported and rotated ice floes. There are several factors that made these observations possible. Kultuk Bay is a relatively narrow region and ice drift is limited , with movement constrained to the westeast direction. Rapid ice deterioration due to thermal melt and limited large-scale wind-driven ice drift also facilitated observation of the eddy's influence on ice transport.

**The power of multi-satellite imagery.** There are still a lot of unknowns in the interpretation of ice cover state from visible, NIR, SWIR, TIR and microwave satellite images and data. When studying natural phenomena a complimentary data

approach is advantageous and it helps to use all existing sources of satellite imagery that may reveal key elements in ice cover development.

While for our case daily or sub-daily frequency of available satellite imagery was sufficient to analyse most of the interesting features, if necessary ice floe tracking can be done on a much shorter time interval. For example, the time interval between subsequent PlanetScope satellites is 90 seconds and this makes it possible to estimate the speed of relatively fast-moving objects, such as river ice (Kääb et al., 2019). In some cases the same area can be covered by non-subsequent tracks making this time lag larger to track slower moving objects.

Satellite radar altimetry provides useful information on the state of the ice cover and the water surface. On 26 April 2020 ice break-up led to the appearance of large areas of open water protected from the wind by drifting ice fields. This calm water acted like a mirror and brought specular reflection and increased Jason-3 backscatter up to 58 dB (Fig 11). Actually the backscatter may have been even higher but was cut out by the ice retracker algorithm at 58 dB. The detached ice floe presented in the northern part of the track on Fig 11 apparently drifted away between 11h 57 min (time of Landsat image) and 20 h (time of Jason-3 observation). The consolidated ice edge remained the same and is resolved by Jason-3 data with high spatial accuracy (290 m). This ability of radar altimetry missions for robust discrimination between open water and ice was noted in our work for T/P-Jason, GFO and ENVISAT-AltiKa series over various Eurasian lakes and rivers (Kouraev et al., 2007; 2008; 2015; Zakharova et al., 2021). However one may also note the spatial variability of backscatter over consolidated ice, such as the decrease of signal over a white (potentially rough and hummocked) ice field (Fig 11, violet dots). This shows the possibility for further ice type classification from radar altimetry data, based on field measurements along satellite tracks and quasi-simultaneous with satellite overpasses. In 2020 the Jason-3 altimetric track was just outside the ring, but in the future, when the altimetric track of some existing or planned such as SWOT (Biancamaria et al., 2016) altimetric mission will pass over an ice ring, it will provide unique data to complement analysis of ice development in the presence of an eddy.

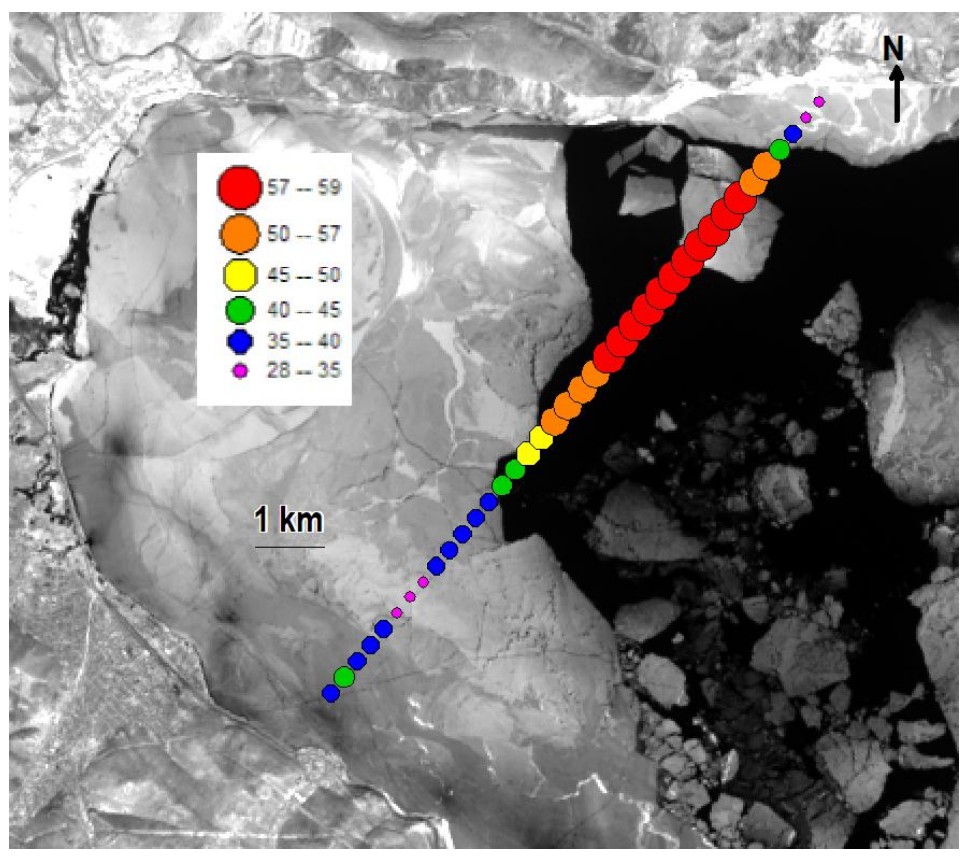


**Figure 11. Backscatter values (dB) for the Jason-3 track 79, cycle 155 (26 April 2020, 20 h local time) overlaid over Landsat-8 image for 26 April 2020, 11h57 local time. Projection UTM 48N.**

Another complementary source of information to assess the interaction of eddies under ice and ice cover itself is thermal infrared imagery. This is less pertinent for the case of ice break-up and warm air over open water. Just after ice melt water

often looks homogeneous, as surface water layer is easily warmed and masks the temperature difference below the surface layer. For example, Landsat 8 TIR image for 26 April 2020 (not shown) reveals warm 3.1-3.6°C open water in the leads near the northern coast. They are warmed due to the effect of sunlight reflected from south-facing mountain slopes on the coast. For the region of the eddy itself the water temperature was 2.4-2.6°C and the thermal contrast was not enough to detect the eddy.


However, during stable ice cover presence there are cases when thermal imagery may reveal the difference in surface temperature due to an underwater eddy, as in winter 2018 near Cape Nizhneye Izgolovye in Middle Baikal (Fig 12, see Fig 3 for geographic location). During our field work on 13-18 February 2018 we took vertical profiles of temperature and identified another lens-like eddy, similar to the ones typical to this region. The thermal image from Landsat 8 on 6 March

2018 shows that increased temperature and heat exchange between the upper dome of the eddy and the ice led to spatial

differences in ice surface temperature. Despite the ice being 60-70 cm thick at the time, the eddy created a circular zone more than 2°C warmer than the surrounding regions. It is important to note, that eddy influence was visible only in the TIR range, and images in the visible, NIR and SWIR ranges for this period did not show any changes making it possible to identify the eddy below.  This eddy did not move during the second field work period in late March, and an ice ring was

formed there in April 2018, this time seen also in the visible and NIR ranges.. Future satellite missions with high-resolution TIR images such as TRISHNA (Roujean et al., 2021), Landsat-9 etc will surely improve temporal coverage and increase the scope of analysis using combinations of thermal and optical images.

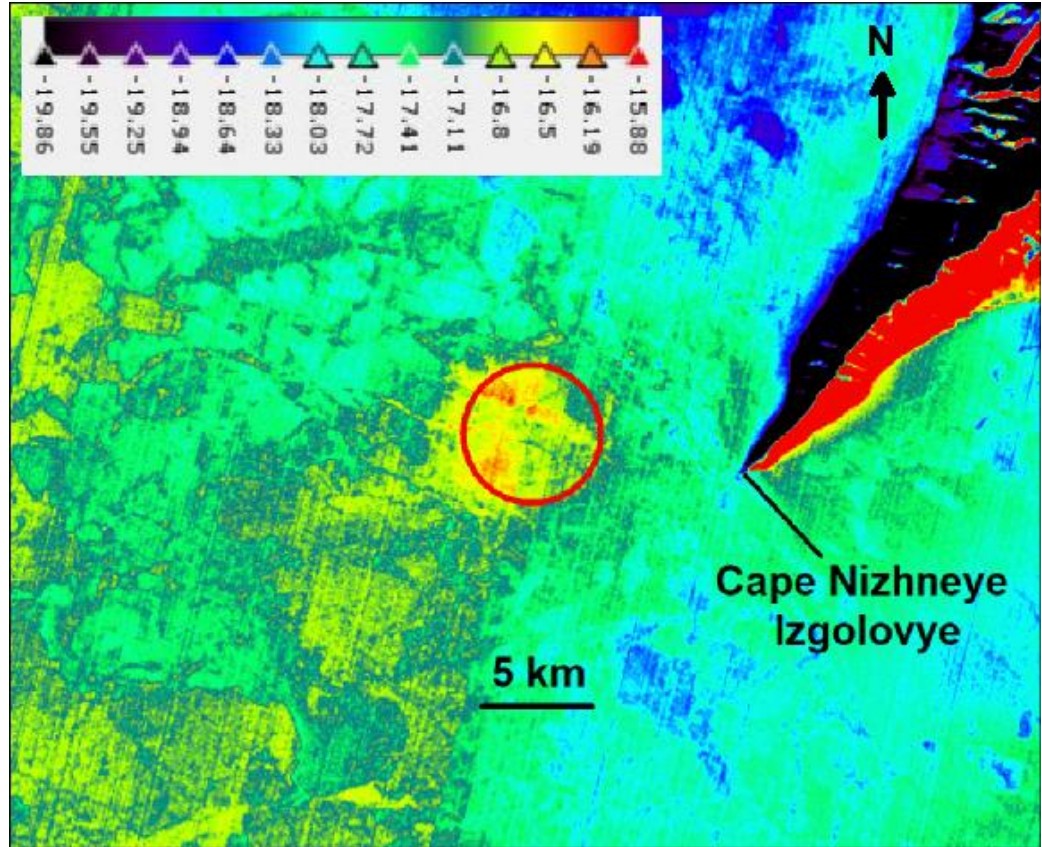

**Figure 12. Ice and land surface temperature (°C) from Landsat 8 image on 6 March 2018. Red circle - position of outer eddy limit as defined from in situ data in February 2018. Projection UTM 48N.**

**Monitoring eddies**. Intrathermocline lens-like eddies are a special type of oceanic eddy. They have been observed in various regions of the World Ocean, but have only recently been discovered in lakes (Kouraev et al., 2016, 2019). While lens-like eddies have a variety of generation mechanisms (Kostianoy and Belkin, 1989), the physics and hydrodynamics of lens-like

eddies in lakes and in the ocean are strikingly similar.

As ice cover is much thinner and weaker in the regions of eddies, the presence of eddies under the ice and the formation of giant ice rings is a clear danger for people travelling on ice in lakes such as Lake Baikal or Lake Hovsgol (Kouraev et al. 2016, 2019). Better understanding of the interaction between eddies and ice, the changes in ice cover in the presence of eddies, timely detection, monitoring and potentially forecast of ice rings or regions of weakened ice, is a major concern for safety on lake ice.


We expect the investigation of the 3-D structure and internal water dynamics of lens-like eddies in lakes to contribute to understanding the same lens-like eddies in the World Ocean. Of particular interest could be study of the eddies in the Arctic Ocean. These eddies are smaller than typical oceanic eddies (Kostianoy, Belkin, 1989) - about 10 km wide - and thus more difficult to detect and explore. In cases when such eddies interact with sea ice , they may create ice rings or similar deformations in ice structure and can be  detected from satellites. The methodology of eddy studies from lake ice may be applied to these eddies.


Investigations of lens-like eddies in the ocean is complicated by the fact that their detection is fortuitous, requires research vessels, deep CTD stations, etc.  Field studies of eddies in ice-covered lakes are greatly facilitated by the presence of stable ice cover. CTD casts from the ice give a unique opportunity to make measurements with a fine spatial resolution of several hundreds or even tens of meters. It is rarely possible to have the same spatial density of observations from a ship.


Field observations alone often lack large-scale view and repeatability. Ice rings and ice metamorphism in the regions of eddies are a surface manifestation of eddies under the ice. In this respect satellite observation is a very effective method to identify ice rings and thus to detect lens-like eddies. This helps to focus field research, as well as to find new eddies. Satellite monitoring provides statistics on the locations, lifetime, and behaviour of ice rings and lens-like eddies.


Further research with the use of multi-satellite imagery, in situ measurements, and numerical and laboratory modelling will bring further information on eddies under ice, their influence on ice cover development, their role in horizontal and vertical heat and mass exchange, their impact on chemistry and biology of the lakes and on human activity.


**Author contribution**

AVK performed the analysis, visualisation and writing the original draft. All authors contributed to the ideas, investigation and analysis, writing and editing the paper.


**Competing interests:**

The authors declare that they have no conflict of interest.

**Acknowledgements.**

We would like to thank A.I. Beketov (Ust' Barguzin, Russia) for helpful discussions on ice metamorphism. We acknowledge
the use of imagery from the NASA Worldview application (https://worldview.earthdata.nasa.gov), part of the NASA Earth
Observing System Data and Information System (EOSDIS). Sentinel-1 and Sentinel-2 images - Modified Copernicus
Sentinel data [2019-2020]/Sentinel Hub. Landsat-8 image courtesy of the U.S. Geological Survey. Jason-3 data were
extracted from the geophysical research data records (GDR) distributed by AVISO+ data portal (avisoftp.cnes.fr).

**Financial support**

This research has been supported by the CNES TOSCA LakeIce, LAKEDDIES and TRISHNA, ESA CCI+ Lakes, CNRS-
Russia IRN TTS projects. A.G. Kostianoy was partially supported in the framework of the P.P. Shirshov Institute of
Oceanology RAS (Russian Academy of Sciences) budgetary financing (Project N 0128-2021-0002). E. Zakharova was
partially supported by the Federal Order № 0147-2019-0001 to Water Problems Institute RAS .

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
