# Peer review of "Giant ice rings in Southern Baikal: multi-satellite data help to study ice cover dynamics and eddies under ice"

_The Cryosphere, 2021_

## Author Response (AR1)

First of all we would like to thank you for the time and attention dedicated to the manuscript and for many constructive remarks.

Our item-by-item answers below are marked ">>". We have already prepared a new version of the manuscript  to to address your suggestions and comments.

On behalf of all co-authors, sincerely

Alexei Kouraev, Toulouse, 27 July 2021

RC1: 'Comment on tc-2021-146', Anonymous Referee #1, 14 Jun 2021
GENERAL COMMENTS

Using a wealth of different types of remote sensing imagery and meteorological data, Kouraev and colleagues provide a fascinating description of the development and disappearance of an ice ring in southern L. Baikal in early spring, 2020. In addition, the authors cleverly estimate the rotational velocity of the eddy current that created the ice ring by determining the rotational speed of ice floes moved by this current. The authors also describe changes in the visual features of the ice cover's surface as weather conditions shift suddenly. With this paper, Kouraev and his colleagues not only extend and strengthen their previous work on the origin of ice rings in lakes, but they also provide excellent examples of how ice clarity and reflectance can change suddenly. Perhaps most importantly, however, their work demonstrates how the visible features of the ice surface on lakes are determined both by meteorological events occurring above the ice and water movements and heat flux occurring from below. This work will be useful not only to cryospheric scientists, but also to hydrodynamicists studying under-ice movement of eddies (or meddies) and to limnologists interested in measuring and predicting primary production under lake ice. The article is well illustrated and written in such a way that it is accessible to scientists from multiple sub-fields. This is no small feat given the highly technical nature of the fields of remote sensing and hydrodynamics.

>> We highly appreciate your positive evaluation of our work, thank you very much!

My main suggestion is to clarify figure and table captions by providing additional information (see below).

SPECIFIC COMMENTS

--A more precise title might be: "Giant ice rings in southern Lake Baikal: multi-satellite data facilitate an analysis of ice cover dynamics and eddies under the ice"

In the alternative title, the word 'evolution' is replaced with 'dynamics', and I encourage the authors to use alternative words such as 'dynamics' or 'development' instead of 'evolution' throughout the entire paper. For many scientists, the word, evolution, implies genetically-based inheritance, so replacing it with more appropriate terminology would increase clarity.

>> Good point. We have made changes to the manuscript (title and elsewhere).

Figure and table captions would benefit from the inclusion of additional information such that all figures and tables are interpretable without reading the main text of the paper. (See specific suggestions below).
>>Done

Fig. 1 – Describe in caption where zero (on x-axis) is located relative to the structure of the ice ring. Also, mention that water temperature data was collected under the ice because the ice is not easily seen in this figure.
>> Done

Fig. 2 – Briefly explain in caption why the ice ring disappears on 20 April and why that image is so dark. Without a brief explanation, the reader may assume problems occurred with the reproduction of the image.
>> Done

Fig. 3 – Excellent figure showing locations of ice rings in five different years in Kultuk Bay.--Place compass rose or a vertical line indicating the direction of north on the figure so the reader can distinguish easily the north from the south coast.--Consider indicating in the figure the position of Tunka Valley.--Explain significance of red dashed line (Jason-3 track No 79). Unclear why this dashed line is in the figure.
>> Tunka valley and compass rose - done. Jason track is referred to Fig 3 in Section 1.3; we also added there that "Radar altimetry do not provide images, but point measurements along the satellite track". We also indicate in this figure location of the Nizhneye Izgolovye Cape (see your later comment).

Table 1 – Date format in footnote should read (DD/MM); not the reverse.Caption – Consider beginning caption with "Inventory of all ice rings…." because I believe this table includes all ice rings observed to date for Kultuk Bay.
>>Done; caption changed.

Figure 4 – Define or explain 'cycle number' in caption. In text of paper and/or this figure caption, you may want to explain that the backscatter coefficient is not synonymous with albedo. Also consider explaining in the caption what these backscatter values are measuring or signify.
>> X axis values modified, now it is just time with dates. Caption modified. We have now provided in the beginning of Section 3 a description of various ice types, and how different surface types are seen on satellite images.

Figures 5 & 7 – Consider combining them into a single figure.
>>We would prefer to keep them separate, as Figure 5 shows the initial state, before ice break-up started in the eddy/ice ring region. Additional issue is that combining them will reduce their size and thus readability. Please also note that now we show Jason-3 data overlaid on satellite image on Figure 5a.

--Indicate in each of the figure captions for Figs. 5,7,9, & 11 that these are satellite images to distinguish these images from Fig. 8 which is a photo shot while standing on the ice and looking down at it. Also, consider marking where the direction north is in these figures so that the compass directions mentioned in the text are more readily interpreted in these images.
>>Done. We put "satellite images" on caption for Figure 5 (Figures 7,9, and 11 have captions "same as Fig. XX" so it should be self-explanatory) and "Photo shots" for Fig. 8 caption.

--Fig. 5 – Explain in caption what the white streaks are. Ice cracks? They are especially visible on 20 April.
>>Done. We now also provide new text in the beginning of section 3 on how different surface types are seen on satellite images.

--Fig. 6 – Very good display of meteorological data, but why doesn't this graph begin April 8th when the ice ring was first detected? I believe this figure begins on April 15 to demonstrate how the ice ring changed markedly in response to weather fluctuations. If that is true, consider stating this in the caption.
>>We added to the figure caption: "Period is selected to represent meteorological conditions for satellite imagery presented in the paper"

--Fig. 7 – Are black areas open water in this figure, Fig. 9, and Fig. 11? Explain in caption.
>>Done .

--Fig. 8 – Indicate in caption that that these photos were taken while standing on the ice as opposed to having been taken from a satellite.--In caption, does the word 'limits' refer to edges? If so, replace 'limits' with 'edges' or 'boundaries'.--Last sentence of caption is redundant with information presented earlier in this caption.
>>Done

Fig. 10 – Great depiction of ice floe movement!
>>Thank you!

Table 2 – State in the caption the year (2020) that these data represent. Also, "Total:219" in the column titled 'Angle of ice floe' is confusing. This is not a sum, because the numbers in that column do not total to 219. Instead, "Total:219" appears to be a restatement of the angle of ice flow on 27 April. Consider removing "Total:219".
>>Done

Fig. 11 – Do the black areas in this photo, indicate open water?
>>Please see answer to your comment on Fig 5.

Fig. 12 – State in caption that this image is for an ice ring in the middle basin of L. Baikal. This is necessary because many readers will not be familiar with the location of Cape Nizhneye Izgolovye.-
>>Done, we also point to Figure 3 when now we show the location of Cape Nizhneye Izgolovye)
-Also state in caption that this image depicts ice surface temperatures.
>>Done, "Ice and land surface temperatures" (we also have the Cape in the image, so it is not only ice).

TECHNICAL CORRECTIONS (typing errors).

-- The manuscript would benefit from redactory editing by a native English speaker. The use of articles (i.e., the, a, of) and verb tense need attention. Authors of scientific articles typically write about their study in the past tense but about the findings of others in the present tense.
>>Done

--line 52 – insert the word "Lake" or "L." in front of all lake names throughout the paper.
>>Done

--lines 36 & 37 – A citation or website seems appropriate at the end of these two sentences.
>>Done

--line 60 – Please place '(clockwise)' after the word 'anticylonic'.
>>Done

--line 68 – Define 'Meddies' as Mediterranean eddies.
>>Done

--line 70 – Remove '(anticyclonic)' because it is now explained in line 60.
>>Done

--line 117 – replace 'gap in time' with wording that is more precise such as 'the lack of daily images' or 'days between useable images'
>>Done

--line 121 – Lines 121-125 present a useful road map or outline to the paper. But, the first section of the paper (i.e., data used) is not mentioned. Insert after the word 'first', 'identify the types of satellite imagery analysed and describe the geographical location of the study'.
>>Done

--line 128 – replace ' the large scale' with 'a large spatial scale'
>>Done

--line 156 – acclimatisation of whom or what? The astronauts?
>>Yes, added "their acclimatisation".

--line 158 – replace 'brought' with 'obtained'
>>Done

--line 196 – replace 'earliest observation' with 'first observation of an ice ring'
>>Done

--line 198 – Was this undocumented ice ring in Kultuk Bay or elsewhere? You present ice ring data for Kultuk Bay in 2019 in Table 1 so please clarify if this undocumented ring also occurred in Kultuk Bay.
>>Yes, it was in Kultuk Bay, text changed.

--line 210 – Replace 'head' with 'western end'
>>Changed to "extremities", as, for example, for the Cape Nizhneye Izgolovye it is a north-oriented end with narrow slopes, a kind of underwater canyon.

--line 270 – Consider replacing 'changes' with 'increases'
>>Done

--lines 308-309 – Great analogy!
>>Thank you!

--lines 323-324 – Replace 'brought a possibility….' with 'presented an opportunity to assess the speed of the eddy current by analyzing ice flow movement.'
>>Done, thank you!

--line 326-7 – replace 'looking at ice floes A...' with 'looking at the changing position of ice floes A…'
>>Done

--line 376 – Cite Figure 2 at end of this sentence?

>>Done (citing figure 7)

--line 392 – Replace 'is transporting and rotating' with 'transported and rotated'. Also replace 'such' with 'this'.
>>Done

--line 394 – replace 'to west-east' with 'west to east'
>>Done

--line 416 – what is fast ice?
>>Changed to "consolidated"

Citation: https://doi.org/10.5194/tc-2021-146-RC1

RC2: 'Comment on tc-2021-146', Anonymous Referee #2, 22 Jun 2021  reply
General Comments:

The authors intention is to perform a largely qualitative descriptive analysis of rings seen at the end of the winter season in ice. This has long been a topic of conversation, with hypotheses regarding the formation of these rings running the gamut of scientific theories to preposterous (i.e. gigantic methane bubbles, Lord of the Rings villains, etc). This article presents the use of multiple remote sensing platforms to generate a near-daily and sometimes sub-daily time series of the environmental and physical conditions that influence the patterns of formation and disappearance of the phenomena. The authors present a hypothesis supported by evidence in which a concave eddy of higher temperature water hinders the re-freezing of surface water on ice floes to create a ring that is evidence in visible-optical images.

Overall, the paper is written well with some whimsical portions that could be reigned in a little. The authors present good evidence that supports their hypothesis, from the CTD to the frequent repeat acquisitions to calculate the speed and direction of ice floes prior to wind-influenced break-up.

There are a number of portions of the paper that could be improved upon that will provide clarity to the reader, with examples provided in detail in my Specific Comments. I recommend these sections require some major improvement prior to publication. Generally, it is unclear to the reader regarding the remote sensing data products that are used in the analysis and the dates of acquisition. There are so many platforms that are discussed (MODIS, Sentinel-1, Sentinel-2, PlanetScope), and some that are not used in the analysis for the eddy at all, (i.e. Jason) that the manuscript would be greatly improved with the inclusion of an overall table.
>> Very good point, thank you. We now provide new text in the beginning of section 3 - various ice types found in Lake Baikal, and how different surface types of water, ice and snow are seen on satellite images, as well as new table summarising this. For the dates - we provide dates of satellite images on each figure. Jason-3 data are now also used to analyse eddy influence on ice on 16 April.

When presented, the data section is quite confusing. Many different type of remote sensing datasets are introduced, but I am still unclear what is being tested? It should be explicitly stated in the Introduction or Methods section – and there is no Methods section, only Data and then Results.
>>Section 3 has been significantly extended and modified.

The relative colour stretching without a symbology is a little misleading for the reader as it does not provide the relative scale of reflectance or backscatter. This is an issue because the authors make interpretative comments throughout based on these figures, but do not indicate the intensity of reflectance. For instance, SWIR for Sentinel-2 showed a clear distribution of ice floes in Figure 7D. However, the SWIR had to have the gain increased to 10x to show the contrast between open water and ice. This low reflectance is indicative of high moisture (but the authors need to indicate this).
>>Symbology is now provided. Concerning SWIR and high moisture please see our answer to your comment on lines 271-274..

Specific Comments:

Line 36 – 36: "The temperature, ice cover and water colour…"
This sentence is a little out of place. Why is the key parameters for the GTN-L important for the introduction?
>>Text changed.

Lines 39 – 44: Many of the statements here need to have references.
>>Done

Line 69: "CTD" Define here.
>>Done

Lines 106 – 109: "However in 2020 it was quite different…. White and dark regions (Figure 2)".
These sentences read quite informally – please revise.
>>Done

Lines 116 – 120: Again these lines read extremely informally. In this section of the paper, should it not be discussed what objectives you are looking to address? What are the hypotheses? Even if this is a paper that identifies the ice rings using multiple platforms, this needs to be identified here.
>>Text modified

Lines 121 – 125: The structure of this paragraph is good here, but what is the hypothesis that you are looking to test? Will the satellite monitoring be the major result?
>>Text modified

Lines 230 – 231: "and afterwards influence… to affect the ice state"
The grammar in this sentence needs to be addressed.
>>Done.

Line 234: "the Tunka Valley"
This is not shown in your study site Figure – could you please include?
>>Done (Figure 3)

Line 233: "a constant Kultuk wind"
A Kultuk wind is not defined. What features are characteristic of a Kultuk wind?
>>We have defined it in lines 174-175 (initial numbering): "Strong and persistent wind from Tunka Valley affects most parts of Lake Baikal. All around Lake Baikal people call this wind "Kultuk" in reference to its origins. "

Line 236: "A large ice floe "A" "
Ice floe "A" is not defined until much later in the paper. Please add this into a figure closer to this line, or refer more to the relative location that you are referring to.
>>There is a series of three images that creates this impression. Actually ice floe "A" is discussed just after reference to "Fig 7a, 21 April 2020) and it is presented there.

Figures 2, 3, 5, 7, 9,11: I feel strongly that each of the figure panels that include remote sensing acquisitions needs to include the symbology that show the high and low reflectance, backscatter, etc that is being presented. The authors acknowledge that the colours have been stretched to improve contrast, but not including the symbology can be viewed as misleading. For example, Figure 7D show cases the SWIR from Sentinel-2 on April 23rd. It shows a) a cloud produced by airplane trails, and b) the ice floes present in the image that do not show the influence of the eddy. I re-created this image in Sentinel-2 Playground and to produce something similar I had to push the gain up 10x to a value of 10. It's important for the reader to understand what the reflectance values of the different bands being shown are, especially when using multiple sensors.
>> Color scales are now provided. See also our answer to your comments to lines 317-318.

Lines 271 – 274: "The radar signal is very sensitive… over the Middle Baikal (Kouraev et al., 2015).

You mention that the radar signal is very sensitive to the volume of scatterers in lake ice – while that is true, it has been shown that for freshwater ice the dominant contributor to overall backscatter in SAR is the roughness at the ice-water interface (Gunn et al., 2018; Atwood et al., 2015).

>> Thanks for these references, we have integrated them. However we do not agree that roughness at the ice/water interface is the dominant contributor for *all* types of freshwater ice. These two mentioned papers analyse shallow lakes with smooth ice surface and inclusion of air bubbles in ice from below as influence of methane seepage. In the case of Lake Baikal the situation is different in many respects. This a deep lake, methane seepage is not constant and is present only in the regions of large gas flares or hot underwater sources. Main type of ice is columnar crystalline (black) ice, surface is often hummocked or rough. Most of vertical air bubbles are on top (due to metamorphism and not due to methane seepage) and they appear in spring. Also for Lake Baikal ice dominant feature during ice metamorphism are air channels just below ice surface, and not these bubbles . Our videos under ice do not show significant variability of ice lower surface as compared to ice surface. We now provide new text at the beginning of Section 3 and have modified other relevant parts of the manuscript.

Mind you, the timing of observations that you are presenting here is in the advanced melt stage – meaning that the area has already been wetted from rain and melt – SAR frequencies will not penetrate through the surface snowpack that has high moisture content, which would restrict the potential to view the ring in this circumstance.

>> Snow on Lake Baikal is thin in general (we now provide a description of ice and snow regime in Section 3) and last snow in Kultuk bay disappeared through sublimation already by the end of March 2020. Rain and snow showers from 18-19 April have already infiltrated into ice surface and on 20 April ice surface was dry. So no snowpack, no wet snow, just dry metamorphised ice. Anyway, while surface/volume scattering is an interesting subject, in this paper we limit ourselves to ice floe detection and movement (in the case of Sentinel-1 SAR) and for analysis of temporal and some spatial variability of ice cover for Jason-3.

Atwood, D. K., Gunn, G. E., Roussi, C., Wu, J., Duguay, C., & Sarabandi, K. (2015). Microwave backscatter from Arctic lake ice and polarimetric implications. IEEE Transactions on Geoscience and Remote Sensing, 53(11), 5972-5982.

Gunn, G. E., Duguay, C. R., Atwood, D. K., King, J., & Toose, P. (2018). Observing scattering mechanisms of bubbled freshwater lake ice using polarimetric RADARSAT-2 (C-Band) and UW-Scat (X-and Ku-Bands). IEEE Transactions on Geoscience and Remote Sensing, 56(5), 2887-2903.

Line 275: "There is no Sentinel-SAR available for this period"
Going on Google Earth Engine it appears that there is a scene available on April 25th, cycle 128, Descending, relative orbit 106 (VV/VH). This could be included here, although it appears that it has been included later in the paper to track the different ice floes in your analysis.
>>This text removed. We use SAR image 25 (26 April) later on, but not for metamorphism analysis, as it was in the final break-up phase at this date.

Line 295: "contrasty" – There needs to be a better word for this, or way to phrase. What about "shows more contrast"?
>>Text modified.

Lines 305 – 310: This paragraph is a little bit whimsical, which happens a couple times in the paper. I feel that this could be analogized or better described than a painter/canvas.
>>We believe that this is probably not the worst possible analogy, Referee #1 seems to like it.

Lines 311 – 313: "Negative air temperature"?... 21-23 April 2020 ß What sensor have you used here?
>>Text modified, data are from Kultuk meteorological station.

Line 313: "Below the ice, eddy influence counteracts… formation of a white crust"
How does this happen? The hypothesis needs to be rooted in thermodynamics, or is it more likely that the eddy moving the open water surrounding the floes will result in extended surface wetting, and therefore lower reflectance?
>> Text modified, this is thermodynamics related to eddy influence. Presence of warmer water below the ice combined with stronger currents lead to increased heat exchange. It is hard to imagine how open water may lead to any significant surface wetting of ice floes.

Lines 317 – 318: "The SWIR band with its longer wavelength… in the upper right corner on Fig. 7d."
To obtain this image, the gain has to be cranked up to 10. I suspect that the reflectance intensity is quite low, consistent with the presence of surface water.
>> Our experience (tens of S2 images and over a hundred Landsat-8 and 7 images) for Lake Baikal ice (different years, different months) shows that very low SWIR values are in general typical for ice cover. This applies both to clear ice and snow-covered ice, independent of stage of ice development. Default colouring (such as 95% in case of SNAP software for example) shows contrast for land but all ice looks black or dark; however by stretching the contrast for the lowest values one may get useful information for ice-covered regions.

Line 331: "Then, like a suitcase thrown"
Again, watch the whimsy.
>>Following your suggestions we pulled some weeds and pruned several whimsical branches :-) in the text. However here we would prefer to keep this comparison. We think that a reasonably good analogy here and there may help readers - specialists from various scientific sub-fields and general public - to easier understand the processes discussed (please also see comments of Referee #1 on this subject).

Line 341 and Fig 10: What about April 23rd? You discuss the other dates shown in Figure 10, but not April 23rd.
>>Ice really opened up only starting from 24 April (we note this in the first paragraph of Section 3.2), there is no specific point in analysing eddy influence on ice floes for April 23rd or before. We also note in the last sentence of the Section 3.1, subsection "Wind influence" that "on 21-23 April .. the position of the ice floes did not change much".

Figure 9: "c) 26 April 2020" – This should be April 25th.
>>We note in the caption for Figure 9 that we present here all time as local time, so 25 April 23:55 GMT is already next day (26 April 07:22) local time.

Paragraph, Lines 407 – 419: This paragraph and Figure 11 seem out of place for this analysis/paper. It does not discuss the ring or ice breakup at all. While Jason and SAR are good options to view ice presence, composition, thickness, etc, they weren't used in this case to observe the ring in the ice. I recommend this section be removed from the analysis.
>>We now provide additional analysis of how Jason-3 observe region just outside the ring on 16 April (Fig 5). We have also modified this paragraph and section 1.3.

Lines 424-426: These lines do not support the statement made in Lines 423 = 424. You state that the thermal image data could be helpful, but not in this case. I understand that you're trying to tie in

the next paragraph here, so you could make it consistent as the same paragraph, or you could work these thoughts into the following paragraph briefly.
>> Good point, thank you. Text modified.